# Impact of COVID-19 on feto-maternal and neonatal health in Karachi, Pakistan, A retrospective cohort study

**Syeda Mahjabeen Zehra**[1], **Sadia Parkar**[1], **Zaubina Kazi**[1], **Asma Pethani**[1], **Ayesha Malik**[2], **Adnan Mirza**[1], **Falak Abro**[1], **Hassan Abdul Jabbar**[1], **Ali Faisal Saleem**[1]*

**1** Department of Paediatrics, Aga Khan University Hospital, Karachi, Pakistan, **2** Department of Obstetrics & Gynaecology, Aga Khan University Hospital, Karachi, Pakistan

* ali.saleem@aku.edu

**Data Availability Statement:** Data is uploaded as a supplementary information.

**Funding:** The authors received no specific funding for this work.

## Abstract

Scientific literature suggests that pregnant women are at greater risk of acquiring a more severe form of COVID-19 exposing both mother and child to a higher risk of obstetric and neonatal complications. These include increased hospitalization rates, ICU admissions, or ventilatory support among pregnant women when compared to COVID-19 negative pregnant womenA case-control study was conducted at the Aga Khan University Hospital, Karachi, Pakistan with the objective of evaluating the clinical presentation of COVID-19 in pregnancy and its effect on maternal and neonatal outcomes. Data was retrospectively collected from April 2020 till January 2022 of obstetric patients with COVID-19 positive cases and were compared with COVID-19 negative cases from the same time. A total of 491 women were included in the study, 244 cases and 247 controls. The most common complication amongst cases was gestational diabetes mellitus (n = 59, 24%), followed by gestational hypertension (n = 16, 31.7%), pre-eclampsia (n = 13, 5%) Pre-rupture of membrane (85.7%). Amongst the COVID positive mothers the most common presenting complaints were fever followed by dry cough, headache, and shortness of breath. It was observed that COVID-19 did not result in increased adverse maternal or neonatal outcomes compared to COVID-19 negative mothers.

## Introduction

The coronavirus (COVID-19) outbreak significantly impacted public health globally and has led to many deaths [1]. It spreads from person to person through respiratory droplets and has high infectivity rates [2]. However, the elderly and those with underlying co-morbidities appear more vulnerable to severe adverse outcomes [3]. Studies exploring the adverse events of COVID-19 on the population have reported that pregnant women are at greater risk of acquiring a more severe form of COVID-19 [4]. The inflammatory nature of COVID-19 during pregnancy exposes both women and their fetuses to a higher risk of obstetric complications and potential complications in exposed newborns [5]. Physiological changes during pregnancy significantly impact the immune system, respiratory system, cardiovascular function, and

**Competing interests:** The authors have declared that no competing interests exist.

coagulation [6]. The immune system adapts during pregnancy to allow for the growth of a fetus, resulting in an altered immune response to infections during pregnancy. Therefore, pregnant women's immune characteristics and clinical outcomes are expected to be different from non-pregnant women with COVID-19 infections [7]. Vertical transmission of the virus can be from the mother to the fetus during the antepartum or intrapartum periods or to the neonate during the postpartum period via the placenta in utero, body fluid contact during childbirth, or through direct contact owing to breastfeeding after birth, however, literature does not show any evidence of vertical transmission [8–10].

The Centers for Disease Control and Prevention (CDC) also reported that pregnant women are more likely to suffer from the severe diseases with increased chances of hospitalization, ICU admission or ventilatory support than non-pregnant women of the same age. However, the mortality remains indistinguishable in pregnant and non-pregnant women [11]. Pre-existing health conditions such as pre-eclampsia or diabetes can contribute to a higher risk of developing complications [12].

There are myriad signs and symptoms of COVID-19 varying from mild to very severe form. The clinical manifestation of COVID-19 in pregnant women seems to be similar as in nonpregnant patients. The most common COVID- 19 symptom being fever followed by cough, myalgia, malaise, sore throat, shortness of breath, and gastrointestinal symptoms which occurred in some patients. There were few cases of severe pneumonia requiring mechanical ventilation [13].

Pregnancy and especially pregnancy with comorbidities might impact fetal or neonatal well-being in the presence of COVID-19. To the best of the authors' knowledge available local literature on maternal and perinatal outcomes of pregnant women infected with COVID-19 at the time of writing article is limited at best. The objective of the study was therefore to evaluate the clinical presentation in pregnancy and its effect on maternal and neonatal outcomes. The findings of this study will add to the existing knowledge about the maternal and perinatal outcomes of pregnant women infected with COVID-19.

## Material and methods

A case-control study was conducted at the Aga Khan University Hospital Stadium Road, Karachi, from 1st April 2020, when the first wave of obstetric patients with COVID-19 positive cases was observed, until 31st January 2022. The Ethical Review Committee of Aga Khan University approved the study ERC # 2020-4978-11514.

The Aga Khan University Hospital (AKUH) is a tertiary care hospital in Karachi, Pakistan. It provides 24 hours emergency services, outpatient clinics, and an inpatient facility comprising of 65-bed general pediatric ward, 12-bed neonatal intensive care unit (NICU), and an eight-bed pediatric intensive care unit (PICU). AKUH has a level III NICU (equipped with ten conventional ventilators, two continuous positive airway pressure (CPAP) drivers, and a high-frequency oscillatory ventilator) providing all neonatal services except extracorporeal membrane oxygenation (ECMO) and hemodialysis. The NICU admits approximately 460 neonates annually. Extremely low birth weight neonates comprise 18% of the total admissions.

The study included COVID-19 positive pregnant women admitted from 1st April 2020 to 31st January 2022 and their controls. The controls were women who tested negative for COVID and delivered in the same unit during the same time period as the cases. The maternal-infant dyads comprised of pregnant women who were admitted to labor and delivery during the study time period. Women who were admitted to labor and delivery but did not deliver a baby were excluded as were those who had multiple births.

During COVID-19 pandemic, all pregnant women admitted for delivery were considered suspected of COVID-19 and were admitted to areas of the Hospital dedicated to their care till

their COVID-19 test results. If the women tested positive for COVID-19, they were transferred to the negative pressure rooms. Patients were transferred between the isolation ward and operating room by a negative pressure isolation transfer cabin by staff wearing BSL-3 protective medical equipment. LSCS was performed in negative pressure operating rooms.

Informed consent waiver was taken since the study data were collected retrospectively from medical records of the pregnant women admitted for delivery and the patient's identity was kept anonymous. Maternal characteristics such as maternal age, gestational week, presenting complaints, symptoms, comorbidities, complications during pregnancy were collected. The labor and delivery information included: method of delivery, postpartum bleeding (PPH)), premature rupture of membranes (PROM), placental abruption, shock, septicemia, asphyxia. Adverse obstetrical and noenatal outcomes included preterm delivery, antepartum hemorrhage, emergency lower segment caesarean, NICU admission, respiratory distress, Apgar score, small for gestation age, sepsis, low birth weight (LBW), neonatal hyperbilirubinemia, significant congenital anomalies and meconium aspiration.

Data were analyzed on SPSS Version 18. Descriptive analysis was performed by generating frequencies and percentages for categoricalvariables and mean with standard deviation for continuous variables. Fisher's Exact Test was used to compare distribution among the groups where statistical significance was set for p-value $<0.05$.

## Results

We enrolled 244 cases and 247 controls in the study, most of the pregnant women in cases and controls (n = 205 (84.0%) and n = 204 (82.5%)) were aged under 35 years of age. 180 (73.8%) cases were >37 weeks gestational age whereas 169 (68.4%) controls had <37 weeks gestational age. Gravidity and parity between both groups was comparable as was the number of miscarriages.

Patients diagnosed with COVID-19 presented with different symptoms, including cough (n = 17, 7%), fever (n = 13, 5%), shortness of breath (n = 5, 2%), sore throat (n = 3, 1%) and headache (n = 2, 1%) (Table 1).

Moreover, with regards to adverse obstetric outcomes, most newborns of SARS-CoV-2-positive mothers were delivered by emergency LSCS 138 (53.9%), 52 (21%) mothers had premature rupture of membrane, 14 (0.6%) had antepartum hemorrhage whereas 108 (43.9%)

**Table 1. Demographics characteristics of pregnant women.**

| Variables | | Cases (n = 244) | Control (n = 247) |
|---|---|---|---|
| | | n (%) | n (%) |
| Age in years | ≤35 | 205 (84.0) | 204 ((82.5) |
| | >35 | 39 (16) | 43 (17.4) |
| Gestational age at admission(weeks) | <37 Weeks | 64(26.2) | 169(68.4) |
| | >37 weeks | 180(73.8) | 78(31.6) |
| Gravida | Primigravida | 78(32) | 74(30) |
| | Multigravida | 138(56.6) | 131(53.0) |
| | Grand Multigravida | 28(11.5) | 42(17) |
| Parity | Nulliparous | 95(38.9) | 89 (36.0) |
| | Multiparous | 145(59.4) | 151 (61.1) |
| | Grand multiparous | 4(1.6) | 7(2.8) |
| Number of Miscarriages | 1 | 54(22.1) | 56(22.6) |
| | 2–4 | 21(8.6) | 31(12.5) |
| | >4 | 2(0.8) | 2(0.8) |

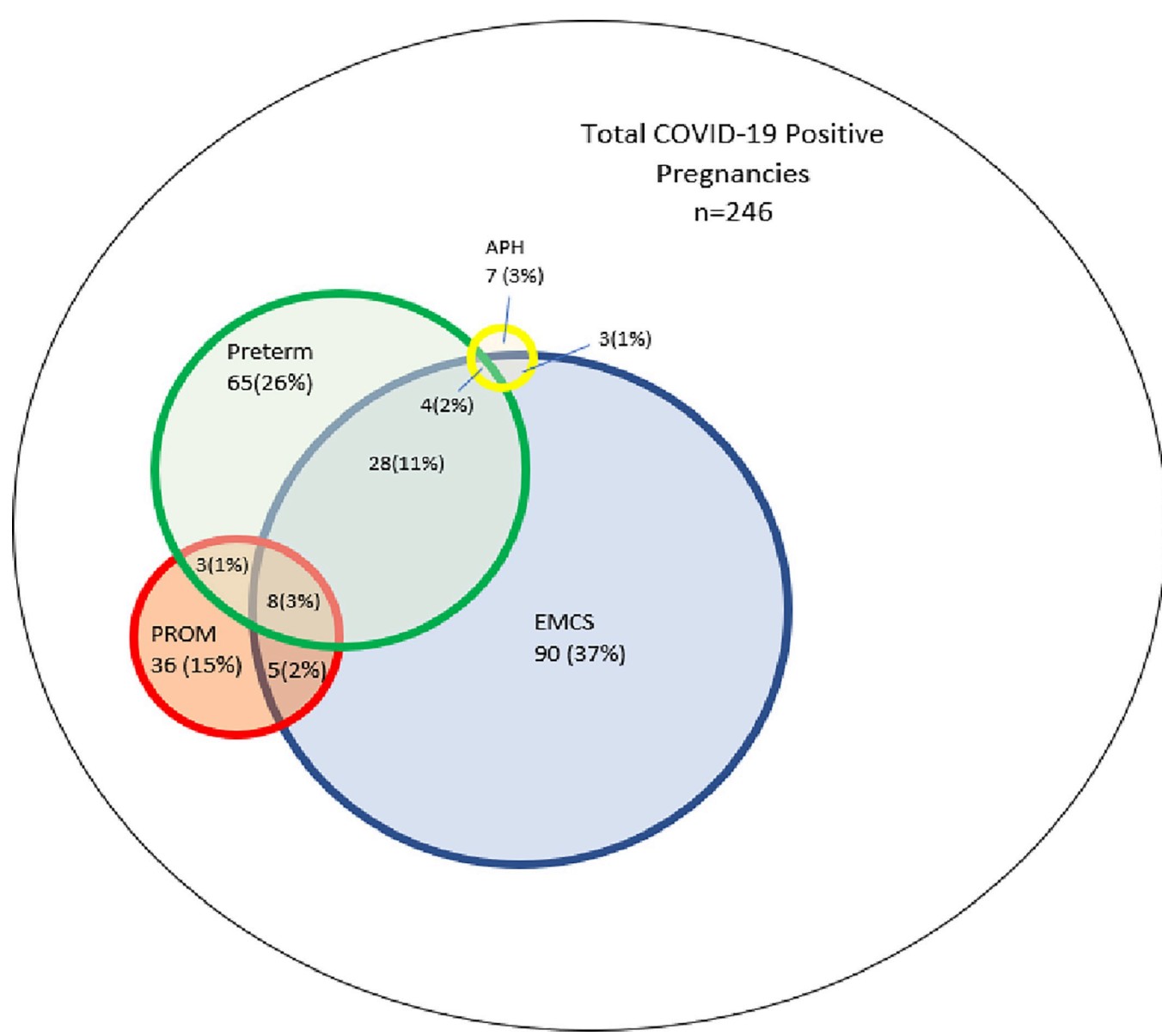

**Fig 1. Detail of maternal outcomes.**

had preterm births. Thirty eight mothers had two events whereas, thirteen mothers were considered as very high risk, they had a combination of at least three events (Fig 1).

Table 2 below details the maternal, neonatal and delivery outcomes in both cases and controls. Amongst the maternal factors, statistical significance association was observed for emergency cesarean section, preterm birth and premature rupture of membranes. The COVID-19 positive women were more likely to have premature rupture of membranes but the cases had significantly more emergency cesarean sections.

Fig 2 shows neonatal outcomes and NICU admission of COVID-19 positive mothers. It can be seen that the frequency of both the adverse neonatal outcomes and NICU admissions was directly proportional to the number of COVID-19 positive mothers. These increased with increased maternal COVID-19 infection during waves: from June to July 2020, December

**Table 2. Maternal and neonatal outcomes in covid positive and negative women.**

| Variables | | | Cases (*n* = 244) | Control (*n* = 247) | OR (95% CI) | p-value | aOR (95%CI) | p-value |
|---|---|---|---|---|---|---|---|---|
| | | | n (%) | n (%) | | | | |
| Maternal Outcomes | Labor | Spontaneous | 102(41.8) | 85(34.4) | 2.69(1.78–4.04) | <0.001 | -------- | ------ |
| | | Induction | 76(31.1) | 14(5.7) | 12.17(6.421–23.077) | <0.001 | 10.22(3.974–26.320) | <0.001 |
| | Emergency Lower Segment Caesarean Section | | 91(37.3) | 177(71.7) | 0.23 (0.160–0.343) | <0.001 | 0.064 (0.019–0.213) | <0.001 |
| | Premature rupture of membrane | | 36(14.8) | 6(2.4) | 6.95(2.87–16.8) | <0.001 | 9.93 (2.735.528) | <0.001 |
| | Preterm birth | | 64(26.2) | 189(76.5) | 0.10(0.72–0.16) | <0.001 | ------ | ------ |
| | Gestational Hypertension | | 16(6.6) | 35(14.2) | 0.425(0.22–0.79) | 0.007 | ------- | ------- |
| | Obstetric cholestasis | | 2(0.8) | 6(2.5) | 3.08(0.61–15.4) | 0.170 | ------ | ------ |
| | Gestational diabetic mellitius | | 59(24.2) | 55(22.3) | 1.11(0.732–1.69) | 0.616 | ------ | ------ |
| | Antepartum Hemorrhage | | 7(2.8) | 6(2.4) | 1.18(0.39–3.58) | 0.762 | ------ | ------ |
| | Preeclampsia | | 8(3.3) | 35(14.2) | 0.2394(0.125–0.455) | <0.001 | ------ | ------ |
| | Types of Delivery | SVD | 106(43.4) | 37(14.9) | 1.64(0.90–3.00) | 0.106 | ------ | ------ |
| | | Emergency LSCS | 91(37.2) | 183(74.0) | 0.28(0.167–0.48) | <0.001 | ------ | ------ |
| | | Elective LSCS | 47(19.2) | 27(11.0) | Ref | | ------ | ------ |
| Neonatal Outcomes | Low birth weight | | 46(18.9) | 157(63.6) | 0.137(0.88–0.20) | <0.001 | ------ | ------ |
| | Small for Gestational age | | 38(15.5) | 159(64.3) | 0.10(0.66–0.15) | <0.001 | 0.153(0.79–0.296) | <0.001 |
| | Respiratory distress | | 16(6.5) | 115(46.5) | 0.08(0.45–0.14) | <0.001 | ------ | ----- |
| | Neonatal Jaundice | | 15(6.1) | 78(31.5) | 0.13(0.77–0.25) | <0.001 | ------ | ------ |
| | Abdnormal Apgar | | 2(0.8) | 85(34.4) | 0.015(0.003–0.64) | <0.001 | 78.86(15.30–376.10) | <0.001 |
| | Presumed Sepsis | | 24(9.8) | 45(18.2) | 0.48(0.28–0.83) | 0.008 | ------ | ---- |
| | NICU admission | | 34(13.9) | 193(78.1) | 0.04(-.02–0.07) | <0.001 | 0.190(0.099–0.362) | <0.001 |

- SVD: Spontaneous labour, LSCS: Lower segment cesarean section.
- Pre-eclampsia is a disorder of pregnancy characterized by the onset of high blood pressure and often a significant amount of protein in the urine.
- Low birth weight: A birth weight of less than 2.5 kilograms. Small for gestational age is a term used to describe a baby who is smaller than the usual number of the weeks of pregnancy.

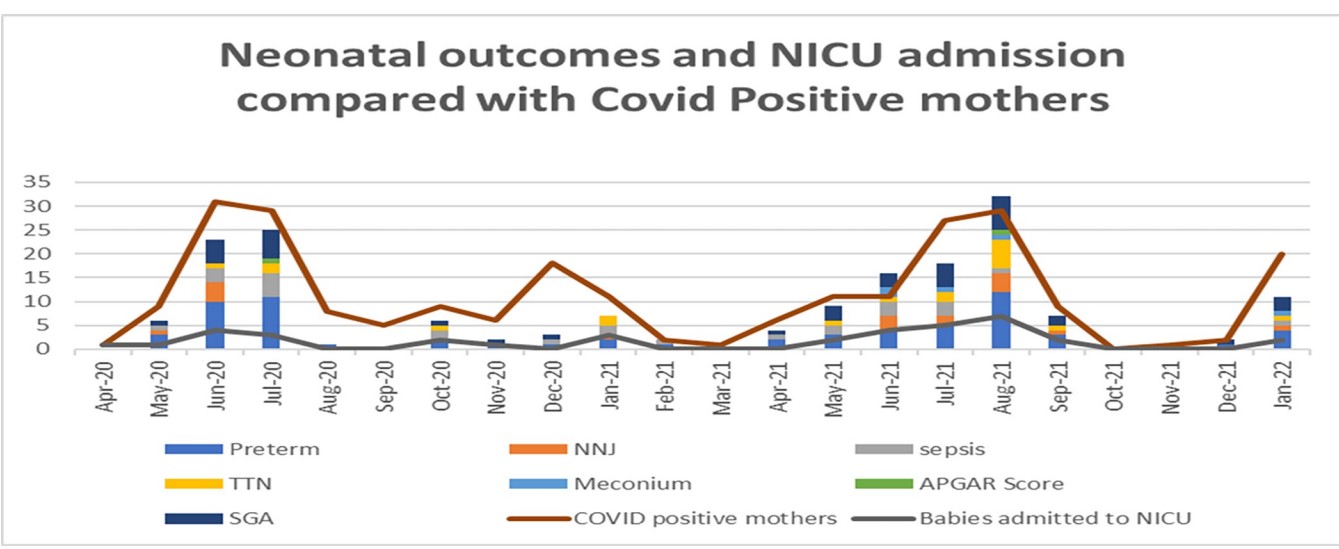

**Fig 2. Neonatal outcomes and NICU Admission compared with COVID positive mothers.**

2020 to January 2021, June to August 2021 and January 2022, following the four waves of COVID-19 infection in Pakistan. This graph also shows that the babies admitted to NICU were mainly SGA and preterm birth.

## Discussion

As mentioned earlier, all women being admitted to labor and delivery were tested for COVID-19, and therefore the PCR test picked up asymptomatic positive cases as well. A previous study showed asymptomatic infection in about a third of pregnant women of whom 12% had preterm delivery, nine neonates were admitted to the intensive care unit, and four of whom were positive for COVID-19, but none developed respiratory symptoms [14].

The most common symptom found in COVID-19-positive mothers was cough followed by fever and dyspnea. Likewise, an earlier study also reported cough to be the most common symptom among pregnant women followed by fever and dyspnea [15].

The study results showed pre-term labor to be among the most common complication of pregnancy among COVID-19 positive mothers, in 25% of cases. An earlier study showed pre-term labor to be more common among COVID-19 positive mothers than those without COVID-19, though the association was insignificant [16]. A recent systematic review and meta-analysis also reported the rate of preterm labor to be higher among pregnant women with COVID-19 (23%),when compared with non-COVID pregnant women [17]. This may be due to an earlier induction of labor in such mothers to avoid COVID-19 complications.

Pre-eclampsia was the third most common co-morbidity amongst the cases in this study. Likewise, a recent study showed that COVID-19-positive women were at higher risk for pre-eclampsia/eclampsia (RR, 1.76; 95% CI, 1.27–2.43) [16]. It has been hypothesized that the hyper-inflammatory state as a result of COVID-19 may lead to hypoxic injury in the placenta resulting in a pre-eclamptic state [18]. In a study related to preeclampsia and COVID-19, pre-eclampsia is independently correlated with COVID-19 during pregnancy (risk ratio [RR], 1.77; 95% confidence interval [CI], 1.25–2.52 in all women, and RR, 1.89; 95% confidence interval [CI], 1.17–3.05 in nulliparous women), and to a lesser extent, gestational hypertension (RR, 1.53; 95% CI, 1.11–2.11). Preeclampsia and COVID-19 were both independently and additively linked to an increased risk of preterm birth, small-for-gestational-age infants, severe perinatal morbidity and mortality [19].

The study results also showed that premature rupture of membrane and preterm labor was the most common maternal complication in COVID-19 positive pregnant females. A previous study showed that this virus could cause complications like premature rupture of membranes and preterm birth in the last trimester of pregnancy [6].

The study results showed that gestational diabetes mellitus is a major comorbidity among suspected covid positive mothers, in 24% of cases. A previous study showed 9.4% of women with covid-19 to have gestational diabetes [20]. Data from the multinational study "INTERCO-VID" study suggests that the pre-existing diabetes mellitus (risk ratio: 1.94; 95% CI: 1.55–2.42), obesity or overweight (risk ratio: 1.20; 95% CI: 1.06–1.37), and gestational diabetes mellitus (risk ratio: 1.21; 95% CI: 0.99–1.46) were all associated to COVID-19. The risk ratio for gestational diabetes mellitus was higher in women who needed insulin, regardless of their weight (risk ratio: 1.79; 95% confidence interval: 1.06–3.01) or whether they were overweight or obese (risk ratio: 1.77; 95% confidence interval: 1.28–2.45) [21]. Regarding mode of delivery, the study results showed that 57% of suspected COVID-19 positive mothers delivered by caesarean section, a finding in line with the published literature. A previous study also showed a majority of such females to have delivered by caesarean section [15,22]. A systematic review reported 80% of mothers with COVID-19 to have had caesarean sections [22]. Another

systematic review recently reported 53% of mothers with signs and symptoms of COVID-19 to have delivered their babies by caesarean section [23]. The reason for these findings could be that due to potential health risks to pregnant COVID-19 positive females, their caregivers might want to avoid any unnecessary medical complications and associated stress during the process of labor.

The study found that nearly all (99%) of the neonates had normal Apgar score at 5 minutes. A previous study reported 97.5% of neonates to have an Apgar score of 5 or more at 5 minutes [15]. This may imply that inspite of the mothers having COVID-19, the neonates weren't affected any more than infants born to healthy mothers.

Literature reports mix findings with regards to pre-term births in COVID-19 positive mothers. The results of this study showed that 27% of COVID-19 positive mothers had pre-term births. An earlier study showed 11% of such mothers to have pre-term births on the other hand a systematic review reported 63.8% of such mothers to have pre-term births [22,23]. Such conflicting results warrant further investigation of this finding before making any meaningful conclusions.

Moreover, our study results further showed that 19% of neonates had low birth weight. An earlier study showed 14.5% of neonates to have low birth weight [15]. A systematic review also reported a large proportion of low birth weight neonates among suspected COVID-19 positive mothers [24]. A higher than expected proportion of low birth weight neonates among suspected COVID-19 positive mothers suggests a negative role of COVID-19 for fetal health and well-being.

The study results also showed that only 15% of neonates were admitted to NICU. A previous study by showed 43% of neonates born to COVID-19 positive mothers to require NICU admission [15]. A systematic review reported 76.9% of such newborns to require NICU admissions [24]. This difference in proportion of NICU admissions may be linked to varied severity of COVID-19 infection among mothers in these studies.

Furthermore, only 1% of neonates in our study died after birth which were a result of neonatal or maternal health complications other than COVID-19. Likewise, an earlier study reported 2% fetal mortality and 0% neonatal mortality in suspected COVID-19 positive mothers [15]. Moreover, a recent systematic review also reported only 0.5% fetal mortality in women with signs and symptoms of COVID-19 [25]. Such low neonatal mortality rates, despite a large proportion of pre-trem births, low birth weight and NICU admissions among the newborns of suspected COVID-19 positive mothers indicate that these adverse neonatal health outcomes of COVID-19 do not necessarily translate into their death.

## Limitations

It is acknowledged that being a single center study, the generalizability of the study findings is limited. This being an observational study, results may be subject to residual confounding factors such as maternal age, previous obstetric history and chronic medical conditions. Moreover, as the study is based on data available from April 2020 till January 2022 may be underpowered to address rare or severe outcomes such as maternal death, ICU admissions and severe pneumonia. An update on the findings reported may be needed as the disease continues to evolve.

## Strength

The study informs clinicians and patients regarding the effect of COVID-19 during pregnancy on maternal and neonatal outcomes.

## Conclusion

Remarkably, it was observed that there was no increased risk of complications in pregnant women who had COVID-19 during pregnancy when compared to non-COVID mothers. This could be because most of the cases were mild and the stusdy findings were from a tertiary care hospital with health care facilities available. While the findings seem promising in light of maternal and fetal outcomes, COVID-19 in itself remains a risk factor. It is recommended that preventive measures especially up to date COVID-19 vaccination with WHO approved vaccines, adequate masking in high risk areas and hand hygiene should be followed by expecting mothers to safeguard them from COVID-19 as it may result in unwanted maternal and neonatal outcomes. In case of being diagnosed with COVID-19, adequate natal care should be ensured for earlier identification of any potential maternal or neonatal complication.

## Supporting information

**S1 Data.**
(SAV)

## Acknowledgments

We are thankful to the staff of Department of Pediatrics and Obstetrics & Gynaecology, AKUH, for their cooperation.

## Author Contributions

**Conceptualization:** Syeda Mahjabeen Zehra, Sadia Parkar, Zaubina Kazi, Asma Pethani, Ayesha Malik, Adnan Mirza.

**Data curation:** Syeda Mahjabeen Zehra, Sadia Parkar, Falak Abro, Hassan Abdul Jabbar.

**Formal analysis:** Syeda Mahjabeen Zehra, Sadia Parkar, Asma Pethani.

**Investigation:** Asma Pethani, Falak Abro, Hassan Abdul Jabbar.

**Methodology:** Syeda Mahjabeen Zehra.

**Project administration:** Zaubina Kazi, Asma Pethani, Ayesha Malik, Ali Faisal Saleem.

**Resources:** Falak Abro, Hassan Abdul Jabbar.

**Software:** Syeda Mahjabeen Zehra, Sadia Parkar.

**Supervision:** Syeda Mahjabeen Zehra, Zaubina Kazi, Asma Pethani, Ayesha Malik, Ali Faisal Saleem.

**Validation:** Hassan Abdul Jabbar.

**Visualization:** Zaubina Kazi, Adnan Mirza, Ali Faisal Saleem.

**Writing – original draft:** Syeda Mahjabeen Zehra, Sadia Parkar.

**Writing – review & editing:** Syeda Mahjabeen Zehra, Sadia Parkar, Asma Pethani, Ayesha Malik, Adnan Mirza, Ali Faisal Saleem.

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
