## [Editor Report · Decision Letter 0]

1 Dec 2022

PGPH-D-22-01826

Impact of COVID-19 on Feto-maternal and Neonatal Health in Karachi, Pakistan, A retrospective cohort study.

Dear Dr. Saleem

Thank you for submitting your manuscript to PLOS Global Public Health. After careful consideration, we feel that it has merit but does not fully meet PLOS Global Public Health’s publication criteria as it currently stands. Therefore, we invite you to submit a revised version of the manuscript that addresses the points raised during the review process.

We look forward to receiving your revised manuscript.

Kind regards,

Babar Tasneem Shaikh, MBBS, MBA, MPH, PhD, FRCP

Academic Editor

Journal Requirements:

2. In the online submission form, you indicated that "Data is a intellectual property of Aga Khan university Hospital and is available on request from the corresponding author.". All PLOS journals now require all data underlying the findings described in their manuscript to be freely available to other researchers, either 1. In a public repository, 2. Within the manuscript itself, or 3. Uploaded as supplementary information.

Additional Editor Comments (if provided):

Please add a list of abbreviations

Mothers enrolled and babies enrolled number is different.

Use one referencing style. Author's names in the Discussion text is Harvard style whereas the rest of the paper uses Vancouver.

Use 'COVID-19' term in standard format.

Limitation do not talk about confounders.

Conclusion is interpretation of results, and does not furnish any recommendations or way forward.

Acknowledgements says thanks to participants; but it was a retrospective study which means you used the hospital data only.
---

## [Decision Letter · Decision Letter 1]

1 Feb 2023

PGPH-D-22-01826R1

Impact of COVID-19 on Feto-maternal and Neonatal Health in Karachi, Pakistan, A retrospective cohort study.

Dear Dr. Saleem,

Thank you for submitting your manuscript to PLOS Global Public Health. After careful consideration, we feel that it has merit but does not fully meet PLOS Global Public Health’s publication criteria as it currently stands. Therefore, we invite you to submit a revised version of the manuscript that addresses the points raised during the review process.

Please submit your revised manuscript by February 15, 2023. If you will need more time than this to complete your revisions, please reply to this message or contact the journal office at globalpubhealth@plos.org. Please include the following items when submitting your revised manuscript:

We look forward to receiving your revised manuscript.

Kind regards,

Babar Tasneem Shaikh, MBBS, MBA, MPH, PhD, FRCP Edin, FRSPH

Academic Editor

Journal Requirements:

Additional Editor Comments (if provided):

Please see the reviewers' comments and revise the paper accordingly with a cover letter stating the changes inserted, edits incorporated and any comments or remarks defended.

Reviewers' comments:

Reviewer's Responses to Questions

**Comments to the Author**

1. If the authors have adequately addressed your comments raised in a previous round of review and you feel that this manuscript is now acceptable for publication, you may indicate that here to bypass the “Comments to the Author” section, enter your conflict of interest statement in the “Confidential to Editor” section, and submit your "Accept" recommendation.

Reviewer #1: (No Response)

Reviewer #2: (No Response)

2. Does this manuscript meet PLOS Global Public Health’s publication criteria? Is the manuscript technically sound, and do the data support the conclusions? The manuscript must describe methodologically and ethically rigorous research with conclusions that are appropriately drawn based on the data presented.

Reviewer #1: Yes

Reviewer #2: Partly

3. Has the statistical analysis been performed appropriately and rigorously?

Reviewer #1: Yes

Reviewer #2: Yes

4. Have the authors made all data underlying the findings in their manuscript fully available (please refer to the Data Availability Statement at the start of the manuscript PDF file)?

Reviewer #1: Yes

Reviewer #2: Yes

5. Is the manuscript presented in an intelligible fashion and written in standard English?

Reviewer #1: Yes

Reviewer #2: No

6. Review Comments to the Author

Reviewer #1: The objective of thic cohort study was to evaluate the clinical presentation in pregnancy and its effect on maternal and neonatal outcomes in a retrospective cohort in a Univerisity Hospital setting in Pakistan. The article is of intertest given the lack of literature in the population of Pakistan. Overall the results are sound and methods adequate, albeit a bit simplistic.

I reccomend a list of contructive criticisms for the authors to immprove the paper before publication.

Major

1. The statistical methods are rather poor and symplistic. Do they have an hystorical cohort before the pandemic to perform a comparisons in order to produce the extent of risk increase due to COVID 19. Thsi may be due easily with bivariate logistic regression adjusting for a handful of major confounder if possible (maternal age, parity,morbidities etc).

2. Can they provide a risk related to severity of COVID 19 for each agnormal obstetric outcomes?

3. The references list is poor and should be enriched and so the discussion. There are important missing references describing association of COVID 19 to gestational diabete mellisus in the discussion (ref 1) and the same can be stated for preeclampsia (2). These references should be used to enrich the discussion of the results in the context of what is already known.

4. The risk of thrombosis after surgery is increased in covid 19 thi salso applies to obstetric ijnterventions (2). Do they have a measure of trhormotic complications? If not disclose this limitation and add the reference to the list along wioth a short note presenting the issue. (ref 3)

5. The quality of figure 1 is so low that I cannot read easily the content. Please reupload it with higher quality/definition

6. Which was the viral variant of concern in Pakistan in that period? What percentage of the population was vaccined in the beginning and end of the study? This information should be added in the discussion according to the available local national literature/data.

7. The strenght of the study is that describes the clinical picture in Pakistan, and there are limited data for this country.

8. In the conclusions I would reccomend to remember that given the high extent of complications in pregnancies affected by CODID 19 vaccine should be reccomended to all pregnant women as well as protection with facial masks.

References

1. Eskenazi B, et al. Diabetes mellitus, maternal adiposity, and insulin-dependent gestational diabetes are associated with COVID-19 in pregnancy: the INTERCOVID study. Am J Obstet Gynecol. 2022 Jul;227(1):74.e1-74.e16. doi: 10.1016/j.ajog.2021.12.032. Epub 2021 Dec 20. PMID: 34942154; PMCID: PMC8686449.

2. Papageorghiou AT, et al. Preeclampsia and COVID-19: results from the INTERCOVID prospective longitudinal study. Am J Obstet Gynecol. 2021 Sep;225(3):289.e1-289.e17. doi: 10.1016/j.ajog.2021.05.014. Epub 2021 Jun 26. PMID: 34187688; PMCID: PMC8233533.

2. COVIDSurg Collaborative; GlobalSurg Collaborative. SARS-CoV-2 infection and venous thromboembolism after surgery: an international prospective cohort study. Anaesthesia. 2022 Jan;77(1):28-39. doi: 10.1111/anae.15563. Epub 2021 Aug 24. PMID: 34428858; PMCID: PMC8652887.

Reviewer #2: Thank you for submitting the manuscript on a topic which is very important and very limited literature is available.

Some of the key observations are as follows although the attachment has them in detail:

The design is a Retrospective Chart Review (RCR), recommended to change it

There are several limitations of RCR which need to be elaborated in the methods/limitation section

The study only uses descriptive analysis - itself a limitation

Referencing need to be corrected/some references seem missing - intext and references section at the end

The manuscript requires native English editor copy editing and proofing

Thanks

7. PLOS authors have the option to publish the peer review history of their article (what does this mean?). If published, this will include your full peer review and any attached files.

**Do you want your identity to be public for this peer review?** For information about this choice, including consent withdrawal, please see our Privacy Policy.

Reviewer #1: No

Reviewer #2: **Yes: **Dr. Syed Khurram Azmat

---

## [Decision Letter · Decision Letter 2]

10 Apr 2023

PGPH-D-22-01826R2

Impact of COVID-19 on Feto-maternal and Neonatal Health in Karachi, Pakistan, A retrospective cohort study.

Dear Dr. Ali

Thank you for submitting your manuscript to PLOS Global Public Health. After careful consideration, we feel that it has merit but does not fully meet PLOS Global Public Health’s publication criteria as it currently stands. Therefore, we invite you to submit a revised version of the manuscript that addresses the points raised during the review process.

We look forward to receiving your revised manuscript.

Kind regards,

Babar Tasneem Shaikh, MBBS, MBA, MPH, PhD, FRCP, FRSPH

Academic Editor

Journal Requirements:

Additional Editor Comments (if provided):

Since paper was revamped and many changes were made for improving it, there came some important observations raised by one of the reviewers. The worthy reviewer feels that his recommendations and suggestions were not catered appropriately, which are in the best interest of the paper. Nevertheless, he has supported the idea to give another chance to the authors to address the short comings in the paper. {Please follow his comments on the second revision}

Reviewers' comments:

Reviewer's Responses to Questions

**Comments to the Author**

1. If the authors have adequately addressed your comments raised in a previous round of review and you feel that this manuscript is now acceptable for publication, you may indicate that here to bypass the “Comments to the Author” section, enter your conflict of interest statement in the “Confidential to Editor” section, and submit your "Accept" recommendation.

Reviewer #1: All comments have been addressed

2. Does this manuscript meet PLOS Global Public Health’s publication criteria? Is the manuscript technically sound, and do the data support the conclusions? The manuscript must describe methodologically and ethically rigorous research with conclusions that are appropriately drawn based on the data presented.

Reviewer #1: Yes

3. Has the statistical analysis been performed appropriately and rigorously?

Reviewer #1: No

4. Have the authors made all data underlying the findings in their manuscript fully available (please refer to the Data Availability Statement at the start of the manuscript PDF file)?

Reviewer #1: Yes

5. Is the manuscript presented in an intelligible fashion and written in standard English?

Reviewer #1: Yes

6. Review Comments to the Author

Reviewer #1: The revision slightly improved the manuscript. However, I noted that some of my previous comments were not considered by the authors such as the discussion of the major evidences emerging from the Intercovid study (ref 1-2) on preeclampsia and gestational diabetes and the discussion on risk of thrombosis (ref 3).

It is interesting to observe that GDM is not significant on their data and this should be discussed in light of other major data with different and significant results (ref 2)

More so the authors unfortunately introduced some flaws and drawbacks with their revision:

Preeclampsia disappeared from the table comparing study groups. Why? This outcome should be added as it is an essential outcome in covid-19 cases (please see ref n 2)..

Moreover, recently, after the first submission, another major evidence emerged showing a significantly improved outcome after COVID 19 vaccine. Women with complete or boosted vaccine doses had reduced risk for severe symptoms, complications, and death (ref 4) Please add in the conclusion a plea for vaccining pregnant women this is essential as a public health intervention! Consider also citing major reference 4.

The conclusion should be restructured: there are several neonatal outcomes which are worst in covid 19 positive as compared to covid 19 negative patients and this does not emerge in the conclusions. More so the study is underpowered to address some rare and severe outcome: maternal death, intensive care unit, severe pneumonia.

Univariable statistics is rather simplistic approach and I recommend multivariable regression model considering some covariate.

Minor

"Pre rupture of membrane (85.7%)" is not sound: premature rupture of membranes

English language should be revised "NICU admission coincided with increased infection...." Unclear... rephrase please.

References

1. Eskenazi B, et al. Diabetes mellitus, maternal adiposity, and insulin-dependent gestational diabetes are associated with COVID-19 in pregnancy: the INTERCOVID study. Am J Obstet Gynecol. 2022 Jul;227(1):74.e1-74.e16. doi: 10.1016/j.ajog.2021.12.032. Epub 2021 Dec 20. PMID: 34942154; PMCID: PMC8686449.

2. Papageorghiou AT, et al. Preeclampsia and COVID-19: results from the INTERCOVID prospective longitudinal study. Am J Obstet Gynecol. 2021 Sep;225(3):289.e1-289.e17. doi: 10.1016/j.ajog.2021.05.014. Epub 2021 Jun 26. PMID: 34187688; PMCID: PMC8233533.

3. COVIDSurg Collaborative; GlobalSurg Collaborative. SARS-CoV-2 infection and venous thromboembolism after surgery: an international prospective cohort study. Anaesthesia. 2022 Jan;77(1):28-39. doi: 10.1111/anae.15563. Epub 2021 Aug 24. PMID: 34428858; PMCID: PMC8652887.

4. Villar J, et al. INTERCOVID-2022 International Consortium. Pregnancy outcomes and vaccine effectiveness during the period of omicron as the variant of concern, INTERCOVID-2022: a multinational, observational study. Lancet. 2023 Feb 11;401(10375):447-457. doi: 10.1016/S0140-6736(22)02467-9. Epub 2023 Jan 17. PMID: 36669520; PMCID: PMC9910845.

7. PLOS authors have the option to publish the peer review history of their article (what does this mean?). If published, this will include your full peer review and any attached files.

**Do you want your identity to be public for this peer review?** For information about this choice, including consent withdrawal, please see our Privacy Policy.

Reviewer #1: No

---

## [Editor Report · Decision Letter 3]

20 Jun 2023

Impact of COVID-19 on Feto-maternal and Neonatal Health in Karachi, Pakistan, A retrospective cohort study.

PGPH-D-22-01826R3

Dear Ali Faisal Saleem,

We are pleased to inform you that your manuscript 'Impact of COVID-19 on Feto-maternal and Neonatal Health in Karachi, Pakistan, A retrospective cohort study.' has been provisionally accepted for publication in PLOS Global Public Health.

Best regards,

Babar Tasneem Shaikh, MBBS, MBA, MPH, PhD, FRCP Edin, FRSPH

Academic Editor